# Functional Ex Vivo Testing of Alveolar Monocytes in Patients with Pneumonia-Related ARDS

**DOI:** 10.3390/cells10123546

**Published:** 2021-12-15

**Authors:** Inès Bendib, Asma Beldi-Ferchiou, Frédéric Schlemmer, Bernard Maitre, Mathieu Surenaud, Sophie Hüe, Guillaume Carteaux, Keyvan Razazi, Jean-Daniel Lelièvre, Yves Lévy, Armand Mekontso Dessap, Véronique Godot, Nicolas de Prost

**Affiliations:** 1Service de Médecine Intensive Réanimation, Hôpitaux Universitaires Henri Mondor, Assistance Publique-Hôpitaux de Paris, CEDEX, Créteil, 94010 Paris, France; ines.bendib@aphp.fr (I.B.); guillaume.carteaux@aphp.fr (G.C.); keyvan.razazi@aphp.fr (K.R.); armand.dessap@aphp.fr (A.M.D.); 2Groupe de Recherche Clinique CARMAS, Faculté de Santé de Créteil, Université Paris Est Créteil, CEDEX, Créteil, 94010 Paris, France; 3INSERM, IMRB, Université Paris Est Créteil, CEDEX, Créteil, 94010 Paris, France; asma.ferchiou@aphp.fr (A.B.-F.); frederic.schlemmer@aphp.fr (F.S.); bm.maitre@gmail.com (B.M.); mathieu.surenaud@inserm.fr (M.S.); sophie.hue@aphp.fr (S.H.); jean-daniel.lelievre@aphp.fr (J.-D.L.); yves.levy@aphp.fr (Y.L.); veronique.godot@gmail.com (V.G.); 4INSERM U955, Equipe 16, Vaccine Research Institute, Créteil, 94000 Paris, France; 5Département d’Hématologie et d’Immunologie Biologiques, Assistance Publique-Hôpitaux de Paris, Groupe Hospitalo-Universitaire Chenevier Mondor, Créteil, 94010 Paris, France; 6Unité de Pneumologie, Service de Médecine Intensive Réanimation, Hôpitaux Universitaires Henri Mondor, Assistance Publique-Hôpitaux de Paris, CEDEX, Créteil, 94010 Paris, France; 7Service d’Immunologie Clinique et Maladies Infectieuses, Hôpitaux Universitaires Henri Mondor, Assistance Publique-Hôpitaux de Paris, CEDEX, Créteil, 9400 Paris, France

**Keywords:** acute respiratory distress syndrome, pneumonia, HLA-DR, TNF, phagocytosis

## Abstract

Biomarkers of disease severity might help with individualizing the management of patients with acute respiratory distress syndrome (ARDS). During sepsis, a sustained decreased expression of the antigen-presenting molecule human leucocyte antigen-DR (HLA-DR) on circulating monocytes is used as a surrogate marker of immune failure. This study aimed at assessing whether HLA-DR expression on alveolar monocytes in the setting of a severe lung infection is associated with their functional alterations. BAL fluid and blood from immunocompetent patients with pneumonia-related ARDS admitted between 2016 and 2018 were isolated in a prospective monocentric study. Alveolar and blood monocytes were immunophenotyped using flow cytometry. Functional tests were performed on alveolar and blood monocytes after in vitro lipopolysaccharide (LPS) stimulation. Phagocytosis activity and intracellular tumor necrosis factor (TNF) production were quantified using fluorochrome-conjugated-specific antibodies. Ten ARDS and seven non-ARDS control patients were included. Patients with pneumonia-related ARDS exhibited significantly lower HLA-DR expression both on circulating (*p* < 0.0001) and alveolar (*p* = 0.0002) monocytes. There was no statistically significant difference observed between patient groups (ARDS vs. non-ARDS) regarding both alveolar and blood monocytes phagocytosis activity. After LPS stimulation, alveolar (*p* = 0.027) and blood (*p* = 0.005) monocytes from pneumonia-related ARDS patients had a significantly lower intracellular TNF expression than non-ARDS patients. Monocytes from pneumonia-related ARDS patients have a deactivated status and an impaired TNF production capacity but display potent phagocytic activity. HLA-DR level expression should not be used as a surrogate marker of the phagocytic activity or the TNF production capacity of alveolar monocytes.

## 1. Introduction

Pulmonary infections account for the vast majority of acute respiratory distress syndrome (ARDS) risk factors [1] and are associated with septic shock in about 70% of cases [2]. Although improvements regarding patient phenotyping have been made using multiparametric approaches combining clinical and biological variables [3,4], no single biomarker obtained from blood samples has been shown to reliably predict the outcomes of patients with pneumonia-related ARDS [5], possibly because of an alveolar compartmentalization of biomarkers during severe pulmonary infections [6].

During sepsis, a sustained decreased expression of the antigen presenting molecule human leucocyte antigen-DR (HLA-DR) on circulating monocytes is used as a surrogate marker of immune failure and higher risk of death [7,8,9]. The overexpression of programmed cell death receptor ligand-1 (PD-L1) also occurs during sepsis and correlates with higher mortality [10]. Monocyte deactivation reflected by down regulation of HLA-DR expression on circulating monocytes [11,12] has been associated with the phenomenon of endotoxin tolerance, defined by a reduction of TNF secretion capacity after ex vivo lipopolysaccharide (LPS) stimulation [13]. The appropriate characterization of sepsis-induced monocytes dysfunctions using phagocytosis or cytokine release assays has become a key field of research for the development of innovative therapeutic strategies. Unfortunately, routine assays provide only measurements of immune cell surface biomarkers [14,15].

Down regulation of HLA-DR on circulating monocytes was consistently associated with an increased risk of nosocomial infections [16] and a higher risk of death [8,16,17,18]. Monitoring circulating monocytes HLA-DR expression has been previously used to identify patients who might benefit from immunomodulatory interventions [19,20]. However, few studies focused on alveolar monocytes despite their central role in the innate frontline defense of pulmonary infections. Contrasting with HLA-DR expression on circulating monocytes, the quantification of HLA-DR on alveolar monocytes was not a prognosis biomarker in patients with pneumonia-related ARDS [6]. This discrepancy led us to question the relationship between alveolar HLA-DR expression level and monocyte functions.

In the current study, which is an ancillary study of the PICARD study [6], we aimed to assess, in a prospective cohort of patients with moderate to severe pneumonia-related ARDS: (1) the phenotype of alveolar and blood monocytes (i.e., levels of HLA-DR and PD-L1 expressions) and compare it to those of non ARDS patients; (2) the function of alveolar and blood monocytes; and (3) explore alveolar monocytes functions according to their level of HLA-DR expression.

## 2. Materials and Methods (Additional Methods Can Be Found in the Appendix A)

### 2.1. Study Design

This was a prospective single-center observational cohort study, which is an ancillary study of the PICARD study [6]. Consecutive patients diagnosed with pneumonia-related ARDS admitted to the medical intensive care unit (ICU) of Henri Mondor Hospital, Créteil, France, from January 2016 to December 2018 were eligible for inclusion in the study.

### 2.2. Patients and Data Collection

All patients with moderate/severe pneumonia-related ARDS [1] were included consecutively with the following inclusion criteria: tracheal intubation and mechanical ventilation for less than 48 h; pulmonary infection diagnosed less than 7 days before ICU admission; bilateral pulmonary infiltrates on chest X-ray; PaO_2_/FiO_2_ ratio ≤ 200 mm Hg with a positive end-expiratory pressure (PEEP) ≥ 5 cm H_2_O. Non-inclusion criteria were as follows: age < 18 years; pregnancy; chronic respiratory failure requiring long-term oxygen therapy; Child-Pugh C liver cirrhosis; lung fibrosis; immunosuppression; SAPS II (Simplified Acute Physiology II score) > 90; irreversible neurological disorders; patients with withholding/withdrawing of life-sustaining therapies and profound hypoxemia (PaO_2_/FiO_2_ < 75 mm Hg).

Control patients, (i.e., non-mechanically ventilated patients free of ARDS, any active infection, diffuse interstitial pneumonia or immunosuppression; *n* = 7) undergoing a bronchoscopy with broncho-alveolar lavage (BAL) and blood sampling as part of routine care, were also included (Appendix A).

Demographics and clinical and laboratory variables were prospectively collected upon ICU admission, at samples collection time points and during ICU stay. Initial severity was assessed using the SAPS II [21] and sequential organ failure assessment (SOFA) scores.

### 2.3. BAL Fluid and Blood Sampling

BAL fluid was collected from all ARDS patients during a bronchoscopy within 48 h of ARDS onset. BAL fluid samples were also collected from control patients. Concomitant heparin anticoagulated blood samples were obtained in ARDS and control patients. Samples were stored at room temperature and analyzed within 2 h.

BAL fluid cytology was performed by direct microscopy after centrifuging broncho-alveolar lavage fluid samples (12,000 revolutions for 10 min) and dying under the May-Grünwald-Giemsa staining. Total (quantified in cells/mL) and differential (i.e., percent of neutrophils, macrophages and lymphocytes) cell counts were measured as recommended [22].

### 2.4. Pretreatment of Monocytes

Directly after collection, blood and BAL samples were added to an equal volume (100 μL/tube) to IMDM (Iscove’s Modified Dulbecco’s Media, Thermo Fisher Scientific, Waltham, MA, USA) with or without 200 ng/mL of *Escherichia coli* LPS (*E. coli* K12 LPS, Invivogen, San Diego, CA, USA), then incubated for 4 h at 37 °C and 5% CO_2_ [23]. Samples drawn to measure the intracellular cytokine synthesis of TNF-α were incubated with a Golgi inhibitor (GolgiStop™ 2 μM, Becton Dickinson, San Jose, CA, USA) for 4 h at 37 °C and 5% CO_2_.

### 2.5. Phagocytosis Assay

After treatment with *E. coli* LPS (200 ng/mL) or medium alone, a fraction of blood and BAL fluid samples were incubated for 30 min with 20 μL of fluorescent *E. coli* particles (fluorescent pHrodo™ Green *E. coli* BioParticles, Thermo Fisher Scientific, Waltham, MA, USA), either at 37 °C and 5% CO_2_ to evaluate the phagocytic activity of cells or at +4 °C to assess fluorescent background for bound but not internalized bacteria. Red cells were then lysed using pHrodo™ BioParticles^®^ Phagocytosis kit for flow cytometry (Thermo Fisher Scientific, Waltham, MA, USA), following the manufacturer’s instructions. Phagocytosis was quantified on both alveolar and circulating monocytes using Fl1/FITC channel and expressed both in percent of positive cells and in delta MFI by retrieving basal MFI of unstained cells.

### 2.6. Immunophenotyping

Blood and BAL fluid surface immunostaining was performed as follows: 100 μL of whole blood or BAL fluid were incubated with the following conjugated-monoclonal antibodies: CD14-ECD/HLADR-PACBLUE/PDL1-BV785 and, for alveolar samples, CD45-AF700 and CD169-AF647 were added to identify alveolar monocytes [24] (BD Biosciences, San Jose, CA, USA; eBioscience, San Diego, CA, USA; or Beckman Coulter, Brea, CA, USA). After washing steps, cells were fixed with PBS 1% PFA and further acquired with a LSRII cytometer (BD Biosciences, San Jose, CA, USA) within 48 h. Flow cytometer analyses were performed with the FlowJo software (version 10; Tree Star, Ashland, OR, USA). HLA-DR and PD-L1 quantification was expressed in percentage of positive monocytes (% positive cells) among the total monocyte population or in delta MFI. Flow cytometry results were expressed both in percentage of positive cells and delta MFI. A difference might be observed between percentage of positive cells and MFI as these provide different information: percentage of positive cells reflects the number of monocytes expressing the biomarker of interest, while MFI is the geometric mean of the biomarker fluorescence on each monocyte.

### 2.7. Assessment of TNF Production by Intracellular Staining

After treatment with *E. coli* LPS (200 ng/mL) or medium alone, blood and BAL fluid samples were immunophenotyped and further used to quantify intracellular TNF synthesis according to the regular PerFix-no centrifuge (nc) (Beckman Coulter, Brea, CA, USA) procedure. Cells were acquired with a LSRII cytometer (BD Biosciences, San Jose, CA, USA) within 48 h. Flow cytometer analyses were performed with the FlowJo software (version 10; Tree Star, Ashland, OR, USA). TNF quantification was expressed in percentage of positive monocytes (% positive cells) among the total monocyte population (positivity threshold was defined with non-stimulated values from the same donor) or in delta MFI.

### 2.8. Gating Strategy

Different gating strategies were used in blood and BAL fluid samples. In BAL fluid samples, and as reported by Yu et al. ([24], we first represented cells on an FSC forward-area (FSC-A) versus CD45 (the leukocyte common antigen) dot blot. The non-granulocyte FSC-A^low^ CD45^+^ cells were further examined on a CD14 versus CD45 dot blot. The subsequent CD14^+^ CD45^+^ cells were then examined in an FSC-A versus CD169 dot blot to distinguish alveolar monocytes (FSC^low^CD14^+^ CD169^−^) from alveolar macrophages (FSC^low^CD14^+^ CD169^+^) (Appendix A). HLA-DR positive (HLA-DR^+)^ monocytes were defined as monocytes with an HLA-DR expression greater than the 99% confidence interval of negative control values (Appendix A). The monocytes CD45^+^, CD14^+^, CD 169^−^ and HLA-DR negative were identified as the monocytes HLA-DR^−^ population in contrast of monocytes CD45^+^, CD14^+^, CD169^−^ and HLA-DR^+^ issued from the same alveolar sample. Circulating monocytes were defined as FSC-A^low^ CD14^+^ cells (Appendix A) in blood samples. Expression of HLA-DR, PD-L1 and TNF on alveolar and blood monocytes was then analyzed.

### 2.9. Inflammation and Endothelium/Alveolar Epithelium Injury Biomarkers Quantification in BAL Fluid Supernatants

Concentrations of 22 biomarkers were measured using Luminex^®^ multiplex bead-based technology (R&D Systems, Minneapolis, MN, USA) in BAL fluid supernatant and serum from 70 pneumonia-related ARDS from the PICARD cohort [6]. These included inflammatory markers and cytokines/chemokines (interleukin (IL)-1Ra, IL-6, IL-7, IL-8, IL-10, IL-12/23p40, IL-13, IL-17A, interferon (IFN)-γ, tumor necrosis factor (TNF)-α, granulocyte-macrophage-colony stimulating factor (GM-CSF), RANTES, IP-10/CXCL10, and Serpin E1), endothelial injury (intercellular adhesion molecule-1 (ICAM-1), vascular endothelial growth factor (VEGF), von Willebrand Factor (vWF), and Angiopoietin (Ang)-1/2) and alveolar epithelium injury (receptor for advanced glycation end products (RAGE), surfactant protein (SP)-D, and amphiregulin). The results of quantification were expressed as fluorescence intensities.

### 2.10. Data Presentation and Statistical Analysis

Continuous variables are reported as median (1st–3rd quartiles) and compared using the Mann-Whitney test. When more than two groups were compared, comparisons were performed using two-way ANOVA with repeated measures, and post-hoc comparisons were performed using the Sidak’s test. Variables are expressed as numbers and percentages and compared with the Chi^2^ or Fischer tests. Spearman correlation coefficients were computed for continuous-continuous variables correlations (Benjamini-Hochberg correction was performed to account for test multiplicity). A *p* value < 0.05 was considered significant. Statistical analyses were performed using GraphPad Prism (version 8.0, GraphPad Software, Inc., San Diego, CA, USA) and R 3.1.2 (The R Foundation for Statistical Computing, Vienna, Austria).

## 3. Results

### 3.1. Patients

One hundred eighty-eight patients with moderate-to-severe pneumonia-related ARDS were admitted to the ICU during the four-year study period, of whom one hundred eighteen had non-inclusion criteria and seventy were included in the PICARD study [6]. Functional tests were performed on alveolar and blood samples from a subgroup of ten patients with pneumonia-related ARDS and a control group of seven non-ARDS control patients (Figure 1). These patients were included in this sub-study consecutively over a six-month period. Among patients with pneumonia-related ARDS (*n* = 10), a microbiological documentation was obtained in eight patients, seven of whom had bacterial infections and one of whom had a viral infection (none of the patients had a bacterial and viral coinfection) (Appendix A).

The comorbidities and clinical and biological characteristics of patients at the time the BAL was sampled (i.e., after a median delay of one day following intubation) are presented in Table 1.

Septic shock was present in 60% of patients (*n* = 6/10) at the time of BAL fluid sampling, and in-hospital mortality was 30% (*n* = 3/10, Table 2). As expected in pneumonia-related ARDS, BAL fluid cellularity was elevated, as compared to non-ARDS controls (median (IQR 25–75): 567 (365–720) vs. 130 × 10^3^/mL (77–137) G/L; *p* = 0.0001) and differential cell counts showed a majority of neutrophils (80.5 (8.8–85.5) vs. 1.0% (0.7–1.0); *p* = 0.0001) and fewer macrophages (median 15.5 (8.8–85.5) vs. 66.0% (66.0–85.5); *p* = 0.001), consistent with alveolar inflammation (Table 1). 

### 3.2. HLA-DR Expression on Alveolar and Blood Monocytes

We quantified the monocytic expression of HLA-DR, a prognostic cell surface biomarker in septic shock patients [17], on alveolar and circulating monocytes. As previously shown [6], within 48 h of tracheal intubation, and as compared with non-ARDS controls, patients with pneumonia-related ARDS exhibited significantly lower HLA-DR expression both on circulating (*p* < 0.0001; Figure 2a,b) and alveolar (*p* = 0.0002 when expressed in MFI; Figure 2b) monocytes. The effect of severe lung infection is thus to decrease not only blood but also alveolar monocytic HLA-DR expression level.

Patients with pneumonia-related ARDS also displayed dramatically higher HLA-DR expression (*p* < 0.0001; Figure 2a,b) on alveolar than on blood monocytes. In non-ARDS patients, we also observed a higher HLA-DR expression on alveolar than on blood monocytes (when expressed in MFI, *p* < 0.0001, Figure 2b), indicating that the alveolar compartmentalization of monocytic HLA-DR expression is not specific to severe lung infections.

In order to further explore the relationship between the HLA-DR expression of alveolar monocytes and the alveolar milieu, we quantified the alveolar concentration of 22 biomarkers and correlated it with monocytic HLA-DR level in a larger cohort of 70 ARDS patients from the PICARD study. We also included in the correlation matrix the clinical parameters of patients. HLA-DR on alveolar monocytes correlated negatively with two clusters: one cluster comprised BAL fluid cellularity and SOFA score, as a reflection of global lung inflammation and organ failures, while the second cluster included cytokines involved in the activation of innate immunity (RANTES, IL-6, IL-1Ra, TNF-α) as well as markers of alveolar epithelial (Amphiregulin, serpin) and endothelial (Ang2) injury (Figure 3).

### 3.3. PD-L1 Expression on Alveolar and Blood Monocytes

As another marker of monocyte dysfunction [25,26], we measured the expression of checkpoint inhibitor and marker of immune suppression, PD-L1, on alveolar and circulating monocytes. PD-L1 expression on monocytes obtained from pneumonia-related ARDS patients was not significantly different from that measured on monocytes of their non-ARDS counterparts (Figure 4a,b). Notably, there was a higher PD-L1 expression on alveolar than on blood monocytes for both pneumonia-related ARDS and non-ARDS control patients (when expressed in MFI, Figure 4b), reflecting an alveolar compartmentalization of monocytic PD-L1 expression.

### 3.4. Functional Testing of Alveolar and Blood Monocytes

Phagocytosis activity of alveolar and circulating monocytes after LPS pretreatment.

The phagocytosis activity of alveolar and blood monocytes was measured before (Figure 5a) and after (Figure 5b) LPS challenge in ten patients with pneumonia-related ARDS and seven non-ARDS controls. There was no statistically significant difference observed between patient groups (ARDS vs. non-ARDS) regarding both alveolar and blood monocytes phagocytosis activity. Yet, as compared to circulating monocytes, alveolar monocytes from ARDS patients exhibited a decreased phagocytosis activity both before (Figure 5c, MFI, *p* = 0.0002) and after (MFI, *p* = 0.0013) LPS challenge (Figure 5d).

Intracellular TNF quantification in alveolar and circulating monocytes after LPS stimulation.

One of the hallmark features of monocyte deactivation is the impaired capacity of TNF production in response to LPS stimulation. We thus assessed the intracellular TNF production of both circulating and alveolar monocytes obtained from pneumonia-related ARDS patients and non-ARDS patients. As expected, LPS challenge ex vivo significantly increased the intracellular TNF synthesis (Figure 6). Such effect was statistically significant in circulating (Figure 6a) and alveolar (Figure 6b) monocytes of non-ARDS patients but not in those of ARDS patients.

When considering intracellular TNF expression after LPS stimulation, alveolar monocytes from pneumonia-related ARDS patients had a significantly lower intracellular TNF expression than non-ARDS patients (Figure 6b,d). Likewise, blood monocytes from ARDS patients showed significantly lower intracellular TNF concentration after LPS challenge than did those from non-ARDS patients (Figure 6c, *p* = 0.005). In addition, alveolar monocytes from pneumonia-related ARDS had a significantly lower LPS-induced TNF expression than circulating monocytes issued from the same patients when expressed in percent of positive cells (*p* = 0.011).

Our finding of a reduced responsiveness to an LPS challenge, illustrated by an impaired capacity of alveolar and blood monocytes from ARDS patients to increase their TNF synthesis capacity, is consistent with an endotoxin tolerant profile.

Phagocytosis activity and TNF production of alveolar monocytes according to their HLA-DR expression.

Finally, we assessed whether the function of monocytes and their level of HLA-DR expression was linked. We thus quantified alveolar monocytes phagocytosis capacity (Appendix A) and LPS-induced intracellular TNF concentrations (Appendix A, results expressed in percentage of positive cells and MFI) according to their HLA-DR^−^ vs. HLA-DR^+^ phenotype and did not observe any statistical difference between these two phenotypes.

## 4. Discussion

The current study assessed the phenotype and function of alveolar and circulating monocytes obtained from patients with pneumonia-related ARDS. The main results of the current study are as follows: (1) Alveolar monocytes from ARDS patients showed a decreased HLA-DR expression, as compared with blood monocytes, and a significant correlation with cytokines involved in the activation of innate immunity, as well as with markers of alveolar epithelial/endothelial injury; alveolar monocytes of ARDS patients, together with those of controls, showed higher PD-L1 expression than those of blood monocytes; (2) The phagocytosis activity and intracellular production of TNF after ex vivo LPS stimulation of alveolar monocytes were decreased in ARDS patients; and (3) Alveolar monocytes phagocytosis capacity and LPS-induced intracellular TNF concentrations did not differ according to their HLA-DR^−^ vs. HLA-DR^+^ phenotype. These findings are consistent with the existence of functional immune alterations in the alveolar compartment of critically ill patients having severe pulmonary infections and ARDS.

Monocytes have a pivotal role in antigen-presentation, together with pathogen clearance and induction and regulation of inflammation. The sequential measurement of HLA-DR expression on circulating monocytes has gained considerable interest over the past several decades in identifying altered immune status among severe ICU patients [27]. However, alveolar monocytes have been poorly investigated, although these represent the first immune lines during pulmonary infections. In the current study, we observed an alveolar compartmentalization of HLA-DR expression in ARDS patients, confirming our previous results [6] and illustrating our cohort is well representative of patients with pneumonia-associated ARDS regarding monocytic HLA-DR expression. Such an expression pattern might not be a consequence of the local activation of monocytes at the site of the primary infection but rather a global effect of inflammation as already reported by Skirecki et al. [28]. Yet, HLA-DR expression on alveolar monocytes obtained from patients with pneumonia-related ARDS was lower than that from non-ARDS patients, consistent with the recruitment of deactivated monocytes in the infected lungs. However, we previously showed that the level of alveolar monocytic HLA-DR expression was not a prognosis biomarker in ARDS patients with severe pneumonia, suggesting that, in the context of severe pulmonary infection, the function of alveolar monocytes might be more informative than their phenotype regarding outcome prediction.

Monocyte PD-L1 overexpression was associated with defects in immune function [29] and independently associated with 28-day mortality in septic shock patients [30]. Genome-wide expression measurements revealed that the expression of PD-L1 and PD-L1/PD-1 pathway-associated genes were significantly upregulated in patients with ARDS who survived or were extubated within 28 days compared to non-survivors or intubated patients [31]. In contrast, blockade of the PD-1 pathway induced ARDS-like pneumonitis in patients during anti-PD-1/PD-L1 therapy [32]. More recently, Monagham et al. reported soluble PD-1 as a potential biomarker in human and experimental extra-pulmonary ARDS [33]. These results suggest that the PD-1/PD-L1 axis is involved in the development of ARDS [3]. In our study, we observed a higher PD-L1 expression on alveolar than on blood monocytes in ARDS and non-ARDS patients, with possible downstream inhibitory effects on T-cell activation. Previous works showed that such a pattern could predispose the host to a dangerous situation upon lung infection [34]. This surface biomarker might be used as a therapeutic target in this setting. Indeed, anti-PD-L1 antibodies have shown benefits in animal models of sepsis and are currently being tested in clinical trials in sepsis [35,36]. Yet, our data show no significant difference in alveolar monocyte expression levels between ARDS patients and controls, suggesting further studies are required to better delineate the involvement of the PD-1/PD-L1 axis in the alveolar compartment at the acute phase of ARDS.

The primary aim of this study was to evaluate alveolar monocyte functions in patients with severe pneumonia and ARDS by exploring their phagocytosis capacity and intracellular TNF production after LPS exposure, together with their relationship with HLA-DR expression. The phagocytic capacity of monocytes was not different in ARDS patients than in controls, contrasting with the downregulation of HLA-DR expression on both alveolar and circulating monocytes, reflecting a defect in their antigen-presenting capacity. This finding was already reported in circulating monocytes from septic patients [37]. Such monocyte pattern resembles that described in alveolar monocytes from cystic fibrosis patients under the effects of pro-resolving mediators, which present enhanced phagocytic activity without evoking proinflammatory responses [15,38]. In ARDS patients, no difference was observed between alveolar and circulating monocytes when the phagocytosis capacity was quantified in percentage of positive cells. In contrast, using MFI, we observed a lower *E. coli*-associated fluorescent signal in alveolar monocytes, which persisted after ex vivo LPS stimulation. It has been previously shown that functional, endotoxin-tolerant monocytes exhibit an increased phagocytic ability coupled with a conserved capacity to kill internalized pathogens, albeit with an impaired antigen-presentation capacity [15,38,39]. Our data suggest that in pneumonia-related ARDS patients, the phagocytosis of alveolar monocytes seems to be less efficient than that of circulating monocytes but as efficient as that of non-ARDS controls.

In the present work, we took advantage of recent intracellular staining flow cytometry methods to detect intracellular TNF production [13] in alveolar and circulating monocytes. As expected, alveolar and circulating monocytes from pneumonia-related ARDS patients were hypo-responsive to ex vivo LPS stimulation. Regarding monocytes, many studies have previously reported the altered capacity of circulating monocytes of septic patients to release proinflammatory cytokines in response to LPS stimulation, TLR agonists, or whole bacteria in vitro [15,40,41]. Fumeaux et al. observed that septic circulating monocytes were characterized by an altered IL-10/TNF intracellular ratio [42]. Accordingly, transcriptomic assays in LPS-challenged peripheral blood mononuclear cells from patients facing gram-negative infections retrieved endotoxin tolerance while enhanced anti-microbial activity and tissue repair [43]. Alveolar monocytes from pneumonia-related ARDS patients exhibited a characteristic behavior of cytokine hypo-responsiveness while exhibiting preserved uptake functions, consistent with a “reprogrammed” rather than an “exhausted” phenotype. This modulation represents a state in which the host attempts to control the initial systemic inflammatory response while maintaining control over the infection. This modulation may represent the return to homeostasis in cases of successful antimicrobial therapy and recovery of underlying disease. In contrast, failure to mount a robust inflammatory response may represent a state of immunosuppression in protracted patients.

Furthermore, we observed a negative correlation between HLA-DR expression on alveolar monocytes and the alveolar concentrations of cytokines involved in innate immunity (i.e., IL-6, RANTES, IL-1Ra, and TNF), suggesting that a proinflammatory microenvironment is associated with alveolar monocyte deactivation. As expected, the SOFA score was negatively correlated with HLA-DR expression on alveolar monocytes, illustrating that the number of organ failures was associated with monocyte deactivation in the lungs, as previously shown in circulating monocytes of septic shock patients [16].

Our study certainly has a number of limitations. This is a monocentric study including a homogeneous population of patients with pneumonia-related ARDS, thus limiting its external validity and the generalizability of the findings. The relatively small number of patients included precluded validating our results in an independent validation cohort, and our findings should thus be reconfirmed in a larger population. We included a limited population of non-ARDS controls free of infection and who were younger than patients from the ARDS group (63 vs. 40 years). On one hand, we cannot exclude that these patients might have mild systemic and/or lung inflammation, as these were patients undergoing a bronchoscopy as part of routine care. On the other hand, HLA-DR expression was previously shown to be reduced in elderly patients, possibly contributing to immunosenescence [44]. Therefore, age could have influenced the observed difference in HLA-DR expression between ARDS and non-ARDS patients. In addition, including COVID-19 patients would have been of great interest but was not feasible because of technical and logistical constraints. Therefore, while the results of this study do not point to a definitive association between monocyte phenotype (HLA-DR and PD-L1 expression) and its function, as a hypothesis-generating study, they do provide sufficient evidence to warrant additional research in this area. As there is no clinical sign of immune dysfunction, it is crucial to identify the best biological tools for patients’ stratification according to their immune status. Since it directly measures the ex vivo capacity of a cell population to respond to an immune challenge, functional testing ideally represents the best method to establish the occurrence of immunosuppression. The current results indicate that the examination of the functional capacities of alveolar monocytes might provide new insights into the pathophysiology of severe lung infections and associated ARDS.

## 5. Conclusions

In conclusion, in patients with pneumonia-related ARDS, this study showed that monocytes have a deactivated status and an impaired inflammatory cytokine production capacity but keep similar phagocytic activity to non-ARDS controls. HLA-DR expression level on alveolar monocytes should not be used as a surrogate marker of phagocytic activity or TNF production capacity. Dedicated tests exploring these key roles of monocytes’ functions during severe lung infections might be interesting to better explore the pathophysiology of the disease and identify new prognostic biomarkers.

## Figures and Tables

**Figure 1 cells-10-03546-f001:**
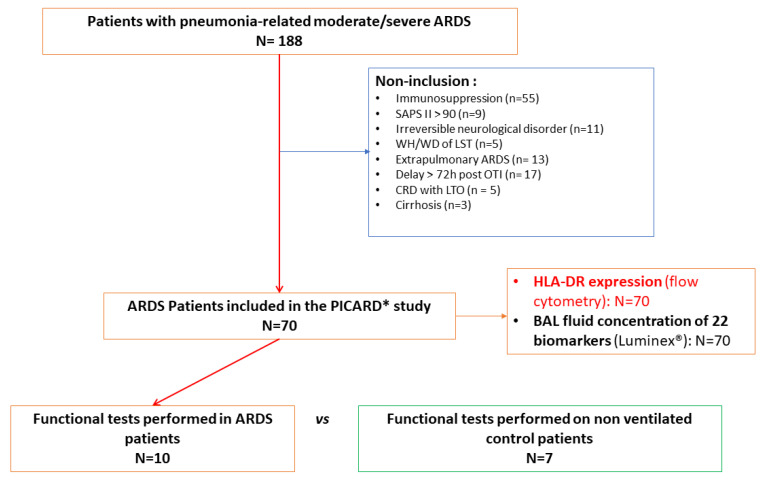
Flow chart of patients with pneumonia-related moderate/severe acute respiratory distress syndrome (ARDS) included in the study. BAL: broncho-alveolar lavage; CRD: chronic respiratory disease; LTO: long term oxygenotherapy; OTI: oro-tracheal intubation; WH/WD of LST: withholding/withdrawal of life-sustaining therapies; * Patients included in the PICARD study are comprehensively described elsewhere [6].

**Figure 2 cells-10-03546-f002:**
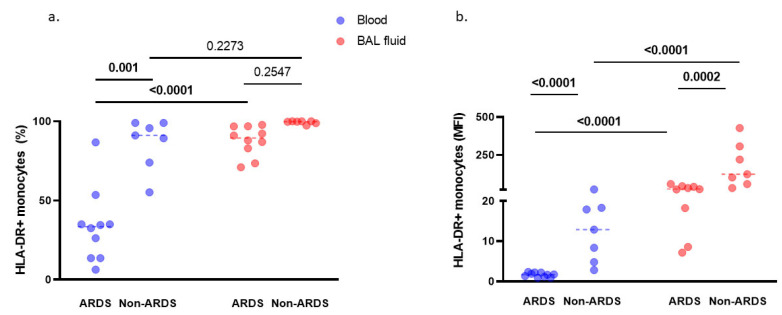
HLA-DR expression on monocytes in broncho-alveolar lavage (BAL) fluid and blood of pneumonia-related ARDS (*n* = 10) and non-ARDS patients (*n* = 7). Expression of monocytic HLA-DR was quantified in percentage of positive cells (**a**) or in mean fluorescence intensity (MFI) (**b**) in standard conditions (medium). By two-way ANOVA with repeated measures, there was a significant effect of group (ARDS vs. non-ARDS, *p* < 0.0001) and of sample compartment (BAL fluid vs. blood, *p* < 0.0001), with significant interaction (group x compartment, *p* < 0.001) on monocytic HLA-DR expression, expressed in percentage of positive cells (**a**). When results were expressed in MFI (**b**), there was a significant effect of group (ARDS vs. non-ARDS patients, *p* = 0.001) and of sample compartment (BAL fluid vs. blood, *p* = 0.0002), with significant interaction (group x compartment, *p* = 0.005). Displayed *p*-values come from post-hoc comparisons performed using the Sidak’s test. Bolded *p* values are significant at the <0.05 level. Horizontal bars represent median values.

**Figure 3 cells-10-03546-f003:**
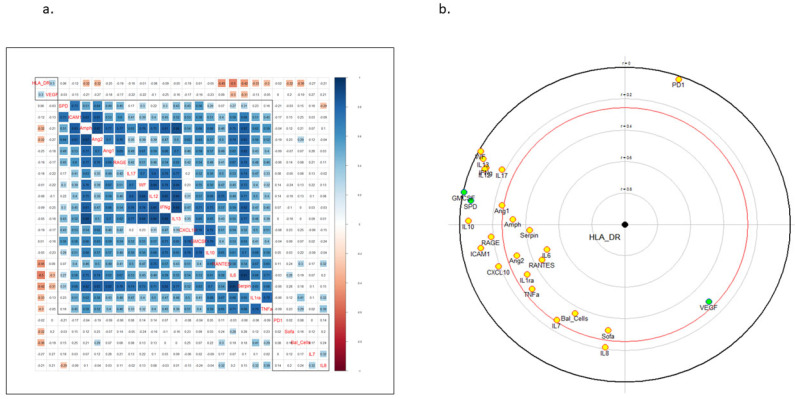
(**a**) Correlation matrix of broncho-alveolar lavage fluid monocytic HLA-DR expression (in MFI), concentrations of cytokines/alveolar epithelial/endothelial injury biomarkers, SOFA score and total BAL cellularity (per mL) in pneumonia-related ARDS patients (*n* = 70). Spearman correlation coefficients are provided for continuous-continuous variables correlations, with positive (blue) and negative (red) correlation coefficients indicating statistical significance at the *p* < 0.05 level after Benjamini-Hochberg correction for test multiplicity. (**b**) Focused principal component analysis (FPCA) displaying the relationships between HLA-DR expression on alveolar monocytes and cytokines/alveolar epithelial/endothelial injury biomarkers concentrations in BAL fluid of pneumonia-related ARDS patients. FPCA is a simple graphical display of correlation structures focusing on a particular dependent variable. The display reflects primarily the correlations between the dependent variable and all other variables (covariates) and secondarily the correlations among the covariates. The dependent variable (HLA-DR expression on alveolar monocytes, expressed in mean fluorescence intensity) is at the center of the diagram, and the distance of this point to a covariate faithfully represents their pairwise Spearman correlation coefficient (using ranked values of continuous variables). Green covariates are positively correlated, and yellow ones are negatively correlated. The red circle delimits statistical significance (at the 5% level). The diagram also shows relationships among covariates. Their relative position is informative as follows: two covariates close to one another indicate a strong positive correlation between them; two diametrically opposed covariates indicate a strong negative correlation between them; two covariates making a right angle with the origin indicate absence of correlation between them.

**Figure 4 cells-10-03546-f004:**
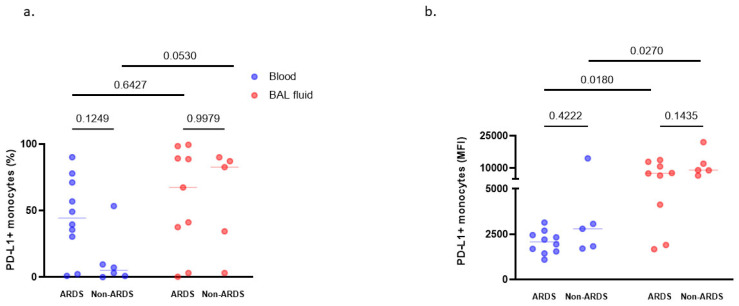
PD-L1 expression on monocytes in broncho-alveolar lavage fluid and blood of pneumonia-related ARDS (*n* = 9) and non-ARDS patients (*n* = 5, anti-PD-L1 antibodies were missing for three experiments). Expression of PD-L1 on monocytes was quantified in percentage of positive cells (**a**) or in mean fluorescence intensity (MFI) (**b**) in standard conditions (medium). By two-way ANOVA with repeated measures, there was a significant effect of group (ARDS vs. non-ARDS, *p* = 0.0490) but no significant effect of compartment (BAL fluid vs. blood, *p* = 0.2484), nor effect of interaction (group x compartment, *p* = 0.2461), when expressed in percentage of positive cells (**a**). When results were expressed in MFI (**b**), there was a significant effect of group (ARDS vs. non-ARDS patients, *p* = 0.0010), also a significant effect of sample compartment (BAL fluid vs. blood, *p* = 0.0192), without significant interaction (group x compartment, *p* = 0.6032). Displayed *p*-values come from post-hoc comparisons performed using the Sidak’s test. Horizontal bars represent median values.

**Figure 5 cells-10-03546-f005:**
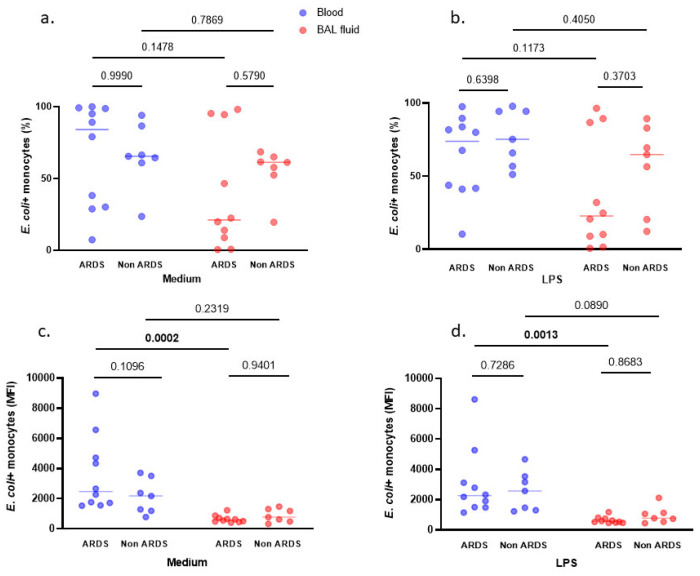
Phagocytosis of *E. coli* particles by alveolar and blood monocytes measured by flow cytometry. The phagocytosis activity of alveolar and blood monocytes has been measured with and without (medium condition) LPS challenge and expressed in percentage of positive cells (%) (**a**,**b**) and in mean fluorescence intensity (MFI) (**c**,**d**) in ten pneumonia-related ARDS and seven non-ARDS patients. Phagocytosis activity was analyzed by two-way ANOVA with repeated measures. When results were expressed in %, there was no significant effect of compartment (BAL fluid vs. blood, *p* = 0.1087) and neither significant effect of group (ARDS vs. non-ARDS, *p* = 0.5264) nor significant interaction (group x compartment, *p* = 0.4903) in medium condition (**a**). After LPS challenge (**b**), there was a significant effect of compartment (BAL fluid vs. blood, *p* = 0.0361) but there was no significant effect of group (ARDS vs. non-ARDS patients, *p* = 0.1397), nor significant interaction (group x compartment, *p* = 0.7593). When results were expressed in MFI, there was a significant effect of compartment (BAL fluid vs. blood, *p* = 0.0003) but neither significant effect of group (ARDS vs. non-ARDS, *p* = 0.2469) nor significant interaction (group x compartment, *p* = 0.1145) in medium condition (**c**). After LPS challenge (**d**), there still was a significant effect of compartment (BAL fluid vs. blood, *p* = 0.0010) but there was neither significant effect of group (ARDS vs. non-ARDS patients, *p* = 0.8645) nor significant interaction (group x compartment, *p* = 0.4072). Displayed *p*-values come from post-hoc comparisons performed using the Sidak’s test. Bolded *p* values are significant at the <0.05 level. Horizontal bars represent median values.

**Figure 6 cells-10-03546-f006:**
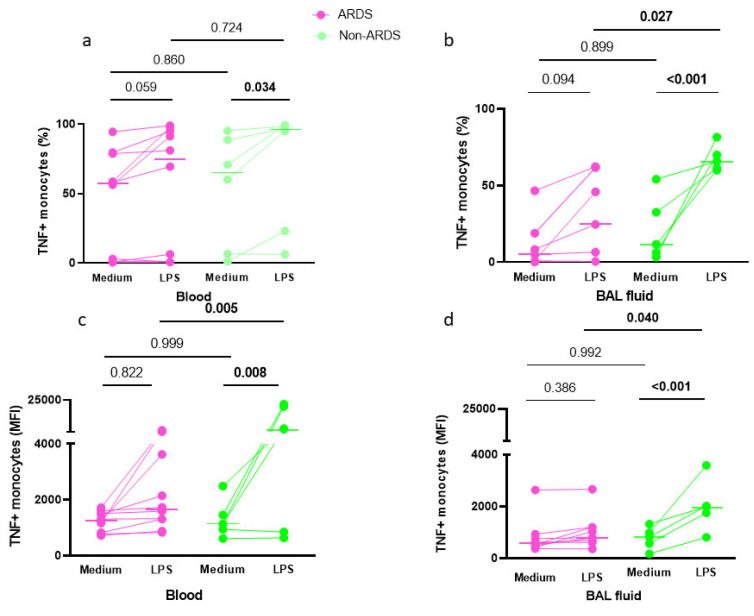
Alveolar and blood monocytes intracellular TNF expression. Intracellular expression of TNF by alveolar and blood monocytes was measured before (medium) and after (LPS) LPS challenge, and expressed in percentage of positive cells (**a**,**b**) and in mean fluorescence intensity (MFI) (**c**,**d**) in ten pneumonia-related ARDS (pink symbols) and seven non-ARDS (green symbols) patients. In blood monocytes (**a**,**c**), by two-way ANOVA with repeated measures, there was a significant effect of the experimental condition (medium vs. LPS, in percentage of positive cells, *p* = 0.003 (**a**) or MFI, *p* = 0.025 (**c**)) and a significant effect of group (ARDS vs. non-ARDS, when expressed in MFI, *p* = 0.005 (c) but not in percentage of positive cells, *p* = 0.547 (**a**)), as well as a significant interaction (experimental condition x group, when expressed in MFI, *p* = 0.027 (c), but not in % of positive cells, *p* = 0.516 (**a**)). In BAL fluid monocytes (**b**,**d**), there was a significant effect of the experimental condition (medium vs. LPS, in percentage (**b**) (*p* < 0.001) and MFI_(**d**) (*p* = 0.015)), no significant effect of group (ARDS vs. non-ARDS), either in percentage (**b**) (*p* = 0.114) or MFI (**d**) (*p* = 0.106), and a significant interaction (experimental condition x group) when expressed in percentage of positive cells (**b**) (*p* = 0.023) but not when expressed in MFI (**d**) (*p* = 0.078). Displayed *p*-values come from post-hoc comparisons performed using the Sidak’s test. Horizontal bars represent median values.

**Table 1 cells-10-03546-t001:** Demographics and clinical and biological features of patients with pneumonia-associated acute respiratory distress syndrome (*n* = 10) and non-ARDS controls (*n* = 7) upon BAL fluid sampling. Continuous variables are presented as median [1st–3rd quartiles]; *p* values come from the Mann-Whitney test; categorical variables are shown as *n* (%); *p* values come from the chi2 or the Fisher exact test, as appropriate. ARDS: acute respiratory distress syndrome; COPD: chronic obstructive disease; WBC: white blood cell. Bolded results are significant at the *p* < 0.05 level.

	All (*n* = 17)	ARDS (*n* = 10)	Control Patients (*n* = 7)	*p*
Age (years)	61 (40–69)	63.5 (58–76)	40 (27–53)	0.11
Male gender	7 (41)	3 (30)	4 (57)	0.95
Comorbidities				
Diabetes mellitus	2 (12)	1 (10)	1 (14)	0.98
COPD	2 (12)	1 (10)	1 (14)	0.98
Chronic heart failure	2 (12)	2 (20)	0 (0)	0.36
Chronic renal disease	2 (12)	2 (20)	0 (0)	0.36
Sickle cells disease	1 6)	1 (10)	0 (0)	0.63
Tobacco	7 (41)	3 (30)	4 (57)	0.58
Blood leukocytes				
WBC counts (G/L)	10.0 (6.0–10.0)	8.5 (5.9–9.7)	6.4 (6.0–9.3)	0.72
Neutrophils (G/L)	7.9 (3.7–12.5)	9.4 (7.9–14.9)	3.7 (3.5–6.3)	**0.03**
Monocytes (G/L)	0.6 (0.4–0.8)	0.8 (0.4–0.9)	0.6 (0.4–0.7)	0.45
Lymphocytes (G/L)	1.3 (0.9–2.0)	0.9 (0.7–0.9)	1.5(1.3–2.3)	**0.02**
BAL fluid cytology				
Leukocytes (10^3^/mL)	250.0 (130.0–570.0)	567.5 (365.0–720.0)	130.0 (77.0–136.6)	**0.0001**
Macrophages (%)	33.0 (15.0–66.0)	15.5 (8.8–85.5)	66.0 (66.0–85.5)	**0.0001**
Neutrophils (%)	60.0 (1.0–81.0)	80.5 (61.2–89.5)	1.0 (0.7–1.0)	**0.0001**
Lymphocytes (%)	6.5 (2–29)	3.2 (2.0–6.1)	30.0 (12.5–33.0)	0.012

**Table 2 cells-10-03546-t002:** Severity and outcomes of patients with pneumonia-associated acute respiratory distress syndrome (*n* = 10). * as defined by the Sepsis 3 definition. Continuous variables are presented as median [1st–3rd quartiles]; categorical variables are shown as *n* (%). ARDS: acute respiratory distress syndrome; Crs: compliance of the respiratory system; ICU: intensive care unit; SOFA: sequential organ failure assessment; SAPS II: simplified acute physiology score II; VFD: ventilator free days.

ARDS (*n* = 10)	N (%) or Median (IQR 25–75)
SAPS II	60 (43–68)
SOFA score	12 (8–14)
ARDS severity	
Moderate	5 (50)
Severe	5 (50)
PaO_2_/FiO_2_ (mm Hg)	96 (72–128)
C_RS_ (mL/cmH_2_O)	26 (22–33)
Septic shock *	6 (60)
Duration of ICU stay (days)	18 (9–33)
Live VFD at day-28 (days)	14 (5–23)
Ventilator-acquired pneumonia	4 (40)
In-hospital mortality	3 (30)

## Data Availability

Contact the corresponding author for direct access to the dataset pertaining to data presented in this article.

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
