# Peer review of "Functional Ex Vivo Testing of Alveolar Monocytes in Patients with Pneumonia-Related ARDS"

_cells, 2021, doi:10.3390/cells10123546_

Round 1
Reviewer 1 Report
In the present study the authors present a study on monocyte function in pneumonia-related ARDS.
- The research hypothesis is interesting and advances the field of sepsis/ARDS research.
- The study is well conducted and well written. In particular the use of samples from bronchoalveolar lavage helps to understand immune processes within pulmonary tissue..
- All control patients suffered from a pulmonary pathology. This seems unavoidable as otherwise bronchoscopy would not be indicated. However, I believe this should be clearly discussed as a limitation within the discussion section of the manuscript, also considering the heterogenicity of the BAL results and the low n-number of the controls.
Author Response
In the present study the authors present a study on monocyte function in pneumonia-related ARDS.
C1: The research hypothesis is interesting and advances the field of sepsis/ARDS research.
C2: The study is well conducted and well written. In particular the use of samples from bronchoalveolar lavage helps to understand immune processes within pulmonary tissue..
R1 and R2: We thank the reviewer for their positive comments on our work as well as for the constructive suggestions made.
C3 : All control patients suffered from a pulmonary pathology. This seems unavoidable as otherwise bronchoscopy would not be indicated. However, I believe this should be clearly discussed as a limitation within the discussion section of the manuscript, also considering the heterogenicity of the BAL results and the low n-number of the controls.
R3: The reviewer indeed raises a very important point. All control patients underwent bronchoscopy and BAL fluid samples to explore radiologic abnormalities or for suspicion of pulmonary tuberculosis because of their geographic origin. They were not admitted in the ICU and none of them was mechanically ventilated or under antibiotic at the time the bronchoscopy was performed. The following sentences have been added to the revised discussion section of the manuscript to address this limitation:
“We included a limited population of non-ARDS controls free of infection and younger than patients from the ARDS group (63.5 vs 40 years old). In one hand, we cannot exclude that these patients might have mild systemic and/or lung inflammation, as these were patients undergoing a bronchoscopy as part of routine care. In the other hand, it was shown that HLA-DR expression was reduced in elderly patients, possibly contributing to Immunosenescence [43]. Therefore, age could have influenced the observed difference in HLA-DR expression between ARDS and non-ARDS patients.”
Reviewer 2 Report
This is an ancillary study of the PICARD study, aiming to assess prospectively :
- The phenotype of alveolar and blood monocytes and compare it with non ARDS patients (HLA-DR expression and PD-L1 expression)
- The function of alveolar and blood monocytes (phagocytosis activity before and after LPS challenge and intracellular TNF concentration before and after LPS challenge)
- The correlation between HLA-DR expression and monocyte functions (on the one hand, the correlation between alveolar concentration of 22 biomarkers and monocytic HLA-DR level on the whole cohort of the 70 ARDS patients of the PICARD study, and on the other hand, the correlation between HLA-DR expression and phagocytosis activity and TNF production in the subset of 10 ARDS patients)
This is an interesting topic and the authors have to be commended to try to address such a difficult question. However, in my opinion, the results and the discussion should be better articulated to answer those 3 questions. Besides, some of the results presented in the current come directly from the PICARD study and bring no new information. The authors should better highlight the main message and the new informations from the current study (which mainly relate to the second objective) and separate them more clearly from the results of the PICARD study. Finally, some results are confusing and seem contradictory ; they need to be better explained.
The first objective was already adressed by the same authors on a larger cohort in a previous study (Bendil et al, Critical Care 2021), showing decreased HLA-DR expression both in the blood and in the alveolar compartment. The only new information regarding monocytes phenotype in the current study is the fact that PD-L1 expression was higher on alveolar monocytes, both in ARDS and control patients, with no significant difference between them.
Regarding their second objective, the authors suggest that phagocytosis of alveolar monocytes was decreased compared to circulating monocytes, in both ARDS patients, before and after LPS challenge, but with no significant difference between ARDS and control patients. Besides, monocytes from ARDS patients showed a decreased intracellular production of TNF after LPS stimulation in the blood and in the alveolar compartment. This is in my opinion the main message of the study and it should be highlighted as so.
The third objective was only partially addressed. Using the full cohort from the PICARD study, the authors show significant correlation between monocytic HLA-DR expression and cytokines involved in the activation of innate immunity, as well as markers of alevolar epithelial/endothelial injury. Again, the message is confusing, because these data seem to have been gathered directly from the PICARD study, the only new information here being the correlation between HLA-DR expression and biomarkers concentrations. This should be better acknowledged. Besides, due to lack of important information (see specific comment below) and to the small size of the groups, it seems difficult to conclude as to a correlation between HLA-DR expression and phagocytosis activity/TNF production.
Beyond these general comments, the authors will find hereunder some specific comments :
Major :
Results
- The patients in the ARDS group were younger than control patients (63.5 vs 40 years old) although it did not reach statistical significance (p=0.11), probably due to the low number of patients. However, it was shown previously that HLA-DR expression was reduced in elderly patients, possibly contributing to immunosenescence (Seidler et al, Age-dependent alterations of monocyte subsets and monocyte-related chemokine pathways in healthy adults, BMC immunology, 2010). Therefore, age could have influenced the observed difference in HLA-DR expression. This discrepancy is a potential source of bias and hinders the interpretation of the observed differences between controls and patients. It should at least be stated and discussed in the limitations.
- Figure 2 shows decreased HLA-DR expression on monocytes in broncho-alveolar lavage fluid and blood of pneumonia-related ARDS vs controls. This brings no new information in comparison with what the authors showed on the full cohort of patients in their previous publication (Bendil et al, Critical Care 2021).
- In Figure 4, the number of patients and the number of points on the graph do not match. For example, there are only 9 points and 5 points for PDL1 expression on monocytes (%) on BAL fluid for ARDS and control patients, respectively (the n is supposed to be 10 and 7, respectively). Could the authors address this discrepancy ? Were some samples removed due to technical issues ?
- Regarding the relationship between phagocytosis activity/TNF production of alveolar monocytes according to their HLA-DR expression, the authors state they did not find any difference between HLA-DR low and HLA-DR high phenotype. The authors do no explain how the HLA-DR low and the HLA-DR high groups were constituted. What was the cut-off value ? Was it based on % of positive cells or on MFI expression ? Were the groups similar in size ? No data can be found in the manuscript. Figure 2 a does not show a significant variability in HLA DR expression among ARDS patients (paraticularly when expressed in MFI). Assuming similar distribution between groups, there would be 5 patients in each group, which is unlikely to be sufficient to observe significant between groups differences. More information is necessary to interpret those results correctly.
- Results expressed in % of positive cells and in MFI are not always consistent (and sometimes show opposite results, such as the expression of PD-L1 monocytes in the blood). This should be stated and discussed in the limitations.
Discussion.
- As stated above, the main results in the discussion do not match the main objectives stated at the end of the introduction. The authors should re-articulate the main results in order to match these objectives.
- Results regarding phagocytosis capacity are contradictory and should be better discussed. On the one hand, the authors suggest that phagocytosis activity of alveolar monocytes was decreased compared to circulating monocytes only in ARDS patients. On the other hand, they state (lines 459-460) « when compared with non ARDS-patients the pagocytic capacity of monocytes was preserved ». Further (line 471), they state : the expression of phagocytosis activity in MFI might be difficult to interpret (…). What do the authors mean ? Is the lower phagocytic activity (in MFI) in alveolar monocytes (compared to circulating) to be interpreted with caution ? However, the authors present it as one of the main results (lines 416-418). This discrepancy and this confusing message needs to be better explained.
Conclusion :
- The authors should be more cautious in their conclusions (lines 514-516 : this study showed that 514 monocytes have a deactivated status and an impaired inflammatory cytokine production 515 capacity but keep potent phagocytic activity) . This is based on limited observations on very few patients and is to be considered as hypothesis-generating. Again, I remain confused about the message regarding phagocytic activity (« keep potent phagocytic activity »).
Minor comments :
In the introduction (lines 50-51), «Pulmonary infections account for the vast majority of ARDS risk factors » is probably overstated, as it is generaly admitted that pneumonia accounts for 40-50% of ARDS etiologies. This sentence should be rephrased.
Author Response
C1- This is an ancillary study of the PICARD study, aiming to assess prospectively:
- The phenotype of alveolar and blood monocytes and compare it with non ARDS patients (HLA-DR expression and PD-L1 expression)
- The function of alveolar and blood monocytes (phagocytosis activity before and after LPS challenge and intracellular TNF concentration before and after LPS challenge)
- The correlation between HLA-DR expression and monocyte functions (on the one hand, the correlation between alveolar concentration of 22 biomarkers and monocytic HLA-DR level on the whole cohort of the 70 ARDS patients of the PICARD study, and on the other hand, the correlation between HLA-DR expression and phagocytosis activity and TNF production in the subset of 10 ARDS patients)
C.2 This is an interesting topic and the authors have to be commended to try to address such a difficult question. However, in my opinion, the results and the discussion should be better articulated to answer those 3 questions. Besides, some of the results presented in the current come directly from the PICARD study and bring no new information. The authors should better highlight the main message and the new informations from the current study (which mainly relate to the second objective) and separate them more clearly from the results of the PICARD study. Finally, some results are confusing and seem contradictory ; they need to be better explained.
C3. The first objective was already adressed by the same authors on a larger cohort in a previous study (Bendib et al, Critical Care 2021), showing decreased HLA-DR expression both in the blood and in the alveolar compartment. The only new information regarding monocytes phenotype in the current study is the fact that PD-L1 expression was higher on alveolar monocytes, both in ARDS and control patients, with no significant difference between them.
R1-R3: we thank the reviewer for their positive comments on our work as well as for the constructive suggestions. Indeed, as pointed out by the reviewer, an alveolar compartmentalization of HLA-DR expression on monocytes issued from ARDS patients was already reported in our previous work. Nonetheless, we believe it is important to show that the patients included in this smaller cohort display the same distribution pattern, although we completely agree with the reviewer that the information pertaining to monocytic HLA-DR expression is not the main message of the article.
As mentioned by the reviewer, PD-L1 expression was significantly higher on alveolar than on circulating monocytes from both ARDS patients and non-ARDS controls, when expressed in MFI, indicating a compartmentalization of this biomarker between blood and alveolar compartments. Previous studies using genome-wide expression measurements revealed that the expression of PD-L1 and PD-L1/PD-1 pathway-associated genes were significantly upregulated in patients with ARDS who survived or were extubated within 28 days compared to non-survivors or intubated patients [1]. In contrast, blockade of the PD-1 pathway induced ARDS-like pneumonitis in patients during anti–PD-1/PD-L1 therapy [2]. More recently, Monagham et al. reported soluble PD-1 as a potential biomarker in human and experimental extrapulmonary ARDS [3]. Yet, our data show no significant difference in alveolar monocyte expression levels between ARDS patients and controls, suggesting further studies are required to better delineate the involvement of the PD-1/PD-L1 axis in the alveolar compartment at the acute phase of ARDS.
To better highlight the main message and the new information from the current study concerning PD-L1 results, the following sentences have been added to the revised version of the Discussion section, as follows:
Page 15, paragraph 2:
“The current study assessed the phenotype and function of alveolar and circulating monocytes obtained from patients with pneumonia-related ARDS. The main results of the current study are as follows: 1) Alveolar monocytes from ARDS patients showed a decreased HLA-DR expression and an increased PD-L1 expression, as compared with blood monocytes;
Page 16, paragraph 3:
“ Genome-wide expression measurements revealed that the expression of PD-L1 and PD-L1/PD-1 path-way-associated gene were significantly upregulated in patients with ARDS who survived or were extubated within 28 days compared to non-survivors or intubated patients [32]. In contrast, blockade of the PD-1 pathway induced ARDS-like pneumonitis in patients during anti–PD-1/PD-L1 therapy [33]. More recently, Monagham et al. reported soluble PD-1 as a potential biomarker in human and experimental extrapulmonary ARDS [34]. These results suggest that the PD-1/PD-L1 axis is involved in the development of ARDS [3]. These results suggest that the PD-1/PD-L1 axis is involved in the development of ARDS”.
and
Page 16, end of paragraph 3:
“Yet, our data show no significant difference in alveolar monocyte expression levels between ARDS patients and controls, suggesting further studies are required to better delineate the involvement of the PD-1/PD-L1 axis in the alveolar compartment at the acute phase of ARDS.”
C4. Regarding their second objective, the authors suggest that phagocytosis of alveolar monocytes was decreased compared to circulating monocytes, in both ARDS patients, before and after LPS challenge, but with no significant difference between ARDS and control patients. Besides, monocytes from ARDS patients showed a decreased intracellular production of TNF after LPS stimulation in the blood and in the alveolar compartment. This is in my opinion the main message of the study and it should be highlighted as so.
R4: We thank the reviewer for this very instructive comment. Indeed, when comparing alveolar to circulating monocytes from ARDS patients we observed a decreased phagocytosis capacity (expressed in MFI but not in percentage of positive cells). It is indeed known that tolerant monocytes have a reduced antigen presenting capacity, illustrated by a downregulation of HLA-DR expression, but that they nevertheless keep a potent phagocytosis capacity. This has been observed in circulating monocytes from severe sepsis patients but alveolar monocytes in the context of septic ARDS have been less explored.
The manuscript was revised in the discussion section by rephrasing the second main result as follows (page 15, paragraph 2):
“2) The phagocytosis activity and intracellular production of TNF after ex vivo LPS stimulation of alveolar monocytes were decreased in ARDS patients”;
The paragraph on the phagocytosis capacity of alveolar monocytes has also been revised. Please refer below to response 12 (R12) to comment 12 (C12).
C5. The third objective was only partially addressed. Using the full cohort from the PICARD study, the authors show significant correlation between monocytic HLA-DR expression and cytokines involved in the activation of innate immunity, as well as markers of alveolar epithelial/endothelial injury. Again, the message is confusing, because these data seem to have been gathered directly from the PICARD study, the only new information here being the correlation between HLA-DR expression and biomarkers concentrations. This should be better acknowledged. Besides, due to lack of important information (see specific comment below) and to the small size of the groups, it seems difficult to conclude as to a correlation between HLA-DR expression and phagocytosis activity/TNF production.
R5: We do understand the reviewer’s comment. As suggested by the reviewer, we have better highlighted the results of the correlations performed between alveolar monocytes HLA-DR expression and markers of alveolar epithelial/endothelial injury and SOFA score in 70 patients. These results aim at better depicting the relationship of alveolar HLA-DR expression with other key parameters.
The main results were rephrased to better highlight this result, as follows (page 15, paragraph 2):
« 1) Alveolar monocytes from ARDS patients showed a decreased HLA-DR expression, as compared with blood monocytes, and a significant correlation with cytokines involved in the activation of innate immunity, as well as with markers of alveolar epithelial/endothelial injury.”
In the revised Discussion section, the following paragraph was added to better acknowledge this result (page 17, last paragraph):
“ We observed a negative correlation between HLA-DR expression on alveolar monocytes and the alveolar concentrations of cytokines involved in innate immunity (i.e., IL-6, RANTES, IL-1Ra, and TNF), suggesting that a pro-inflammatory microenvironment is associated with alveolar monocyte deactivation. As expected, the SOFA score was negatively correlated with HLA-DR expression on alveolar monocytes, illustrating that the number of organ failures was associated with monocyte deactivation in the lungs, as previously shown in circulating monocytes of septic shock patients [4]”
Regarding the third objective of the study, we acknowledge that we had not provided enough data in the previous version of the manuscript. Because the number of patients included (n=10) precludes performing correlations (limited statistical power, as pointed out by the reviewer), we separated the monocytes in two groups, i.e., those who expressed HLA-DR (HLA-DR+) and those who did not (HLA-DR-), and compared the phagocytosis activity and TNF intracellular secretion activity between them. We have now provided two additional figures (Figures S2 and S3 in the online supplement) to present these data and have modified the main text accordingly, as follows:
Page 15, first paragraph:
“We thus quantified alveolar monocytes phagocytosis capacity and intracellular TNF concentrations, according to their HLA-DR- vs HLA-DR+ phenotype and did not observe any statistical difference between these two phenotypes: neither the phagocytosis activity (Figure S2) nor the LPS-induced intracellular TNF production were different (Figure S3, results expressed in percentage of positive cells and MFI). »
- C6: The patients in the ARDS group were younger than control patients (63.5 vs 40 years old) although it did not reach statistical significance (p=0.11), probably due to the low number of patients. However, it was shown previously that HLA-DR expression was reduced in elderly patients, possibly contributing to immunosenescence (Seidler et al, Age-dependent alterations of monocyte subsets and monocyte-related chemokine pathways in healthy adults, BMC immunology, 2010). Therefore, age could have influenced the observed difference in HLA-DR expression. This discrepancy is a potential source of bias and hinders the interpretation of the observed differences between controls and patients. It should at least be stated and discussed in the limitations.
R6: The reviewer raises an important point. As suggested, the following additional sentences have been added to the revised “limitations” subsection of the revised version of the manuscript (page 19, second paragraph):
“We included a population of non-ARDS controls free of infection and who were younger than patients from the ARDS group (63 vs 40 years). In one hand, we cannot exclude that these patients might have mild systemic and/or lung inflammation, as these were patients undergoing a bronchoscopy as part of routine care. In the other hand, it was shown that HLA-DR expression was reduced in elderly patients, possibly contributing to immunosenescence [45]. Therefore, age could have influenced the observed difference in HLA-DR expression between ARDS et non-ARDS patients.”
C7: Figure 2 shows decreased HLA-DR expression on monocytes in broncho-alveolar lavage fluid and blood of pneumonia-related ARDS vs controls. This brings no new information in comparison with what the authors showed on the full cohort of patients in their previous publication (Bendib et al, Critical Care 2021).
R7: We understand the point raised by the reviewer. We indeed present monocytic HLA-DR expression in this sub-cohort and show the lung-blood compartmentalization of this biomarker. This information has been already reported by our team [5] and we do acknowledge that these results per se do not bring new insights in the understanding of HLA-DR expression on monocytes in the pathophysiology of ARDS. Yet, we believe it important to demonstrate that our subset of patients (n=10) displays the same pattern of HLA-DR distribution in both compartments than that displayed in the previous PICARD study. In this context, we have reviewed the dedicated paragraph of the Results section (3.2) and the second paragraph of the discussion section, as follows:
“As previously shown [6],… »
“In the current study, we observed an alveolar compartmentalization of HLA-DR expression in ARDS patients, confirming our previous results [6], and illustrating our cohort is well representative of patients with pneumonia-associated ARDS regarding monocytic HLA-DR expression..”
R8: In Figure 4, the number of patients and the number of points on the graph do not match. For example, there are only 9 points and 5 points for PDL1 expression on monocytes (%) on BAL fluid for ARDS and control patients, respectively (the n is supposed to be 10 and 7, respectively). Could the authors address this discrepancy ? Were some samples removed due to technical issues ?
C8 : We thank the reviewer for raising this point. Indeed, only samples from 9 ARDS patients and 5 non-ARDS controls could be analyzed with anti-PD-L1 antibodies for technical reasons (antibodies were not available during experiments concerning our first ARDS patient and the two first non-ARDS controls). This information has now been provided in the legend of Figure 4.
C9: Regarding the relationship between phagocytosis activity/TNF production of alveolar monocytes according to their HLA-DR expression, the authors state they did not find any difference between HLA-DR low and HLA-DR high phenotype. The authors do no explain how the HLA-DR low and the HLA-DR high groups were constituted. What was the cut-off value ? Was it based on % of positive cells or on MFI expression ? Were the groups similar in size ? No data can be found in the manuscript. Figure 2a does not show a significant variability in HLA DR expression among ARDS patients (particularly when expressed in MFI). Assuming similar distribution between groups, there would be 5 patients in each group, which is unlikely to be sufficient to observe significant between groups differences. More information is necessary to interpret those results correctly.
R9: The reviewer indeed points out an important aspect of the Methods section and we apologize if this was not clearly written. The terms HLA-DR “low” and “high” may indeed have been misleading, as we in fact isolated HLA-DR+ and HLA-DR- alveolar monocytes, as further explained below.
Using cell populations within the experiment (tube) as negative controls has been considered by a consensus of cytometrists [6] to be the most appropriate method to quantify non-specific binding by the specific antibody. These cells were part of the experiment from the beginning and were exposed to all steps of processing and staining. In the gating strategy of alveolar monocytes, we represented living singlet cells on a CD14 versus CD45 (the leukocyte common antigen) dot blot. The subsequent CD14+ CD45+ cells were then examined in a forward scatter-area (FSC-A) versus CD169 dot blot to distinguish alveolar monocytes (FSC-AlowCD14+ CD169-) from alveolar macrophages (FSC-AlowCD14+ CD169+) [7]. Then, HLA-DR expression was assessed on alveolar monocytes (Figure S1a). To set up the positive threshold of HLA-DR staining from background, we considered both the autofluorescence of unstained cells and the unspecific staining detected on CD45- CD14- cells (not shown). The percentage of HLA-DR positive monocytes was defined as the percentage of monocytes with an HLA-DR expression greater than the 99% confidence interval of negative control values. The monocytes CD45+, CD14+, CD 169-, HLA-DR- were identified as the monocytes HLA-DR- in contrast of monocytes CD45+, CD14+, CD169-, HLA-DR+ issued from the same sample. For Phrodo/E. coli internalization, we considered the autofluorescence of unstained cells and Phrodo background due to its binding on cell surface. Phrodo background was evaluated on alveolar cells incubated at +4°C. The LPS-induced TNF expression was assessed in alveolar monocytes after a step of permeabilization (Figure S1b). To set the positive threshold of TNF staining, we considered the background observed in unstimulated control cells.
This strategy allowed us to have the same number of samples in each group (i.e., 10 ARDS patients), as we divided every monocyte population issued from each ARDS patient into two sub-populations: HLA-DR- and HLA-DR+ alveolar monocytes.
As suggested by the reviewer, and to better meet the third objective of the article, the raw data of phagocytosis activity and TNF intracellular secretion according to alveolar monocyte HLA-DR expression (+ vs -) are now provided in the Online supplementary material (Figures S2 and S3) and the Results section of the revised manuscript has been modified accordingly.
We have also added in the revised Method section (“2.8. Gating strategy” subsection) how we did determine HLA-DR- (negative) and HLA-DR+ (positive) monocytes among the whole monocyte population issued from each ARDS patients (see the revised subsection “2.Methods”; “2.8 Gating strategy”):
“HLA-DR positive (HLA-DR+) monocytes were defined as monocytes with an HLA-DR expression greater than the 99% confidence interval of negative control values (Figure S1). The monocytes CD45+, CD14+, CD 169-, HLA-DR negative were identified as the monocytes HLA-DR- population in contrast of monocytes CD45+, CD14+, CD169-, HLA-DR+ issued from the same alveolar sample.”
C10 : Results expressed in % of positive cells and in MFI are not always consistent (and sometimes show opposite results, such as the expression of PD-L1 monocytes in the blood). This should be stated and discussed in the limitations.
R10: This is indeed an important point. Flow cytometry results can be expressed either in percentage of positive cells or in geometric mean fluorescence intensity (MFI). In our study, we expressed our results using both percent positive cells and MFI of a population (expressing the biomarker of interest or cytokine producing cells), with both measurements compared to a control population of untreated or unstimulated cells, in order to provide the most comprehensive flow cytometry results, as these variables do not provide the same information: percentage of positive cells reflect the amount of cells expressing the biomarker of interest, while MFI is the geometric mean of the biomarker fluorescence on each cell. It is commonly admitted that a higher percentage of positive cells reflect a higher expression of the biomarker of interest on the whole cells population, comparing to another one. Yet, a higher MFI could be interpreted as a higher expression of a biomarker on a cell population. However, MFI must be interpreted carefully to ensure other factors are not confounding its interpretation. For instance, a larger cell with a larger membrane surface area can appear brighter than a smaller cell of the same type. Thus, it is important to interpret carefully MFI results as cell size or compensation may confound results. To precise this point, the following paragraph in the revised Methods section (“2.6 immunophenotyping”) was added to the revised manuscript, as follows:
“Flow cytometry results were expressed both in percentage of positive cells and delta MFI. A difference might be observed between percentage of positive cells and MFI as these provide different information: percentage of positive cells reflects the amount of monocytes expressing the biomarker of, interest while MFI is the geometric mean of the biomarker fluorescence on each monocyte.”
C11: As stated above, the main results in the discussion do not match the main objectives stated at the end of the introduction. The authors should re-articulate the main results in order to match these objectives.
R11: As suggested by the reviewer, the main results listed at the beginning of the Discussion section have now been re-articulated in order to match the main objectives stated in the introduction.
“The main results of the current study are as follows: 1) Alveolar monocytes from ARDS patients showed a decreased HLA-DR expression, as compared with blood monocytes, and a significant correlation with cytokines involved in the activation of innate immunity, as well as markers of alveolar epithelial/endothelial injury. Alveolar monocytes of ARDS patients, together with those of controls, showed higher PD-L1 expression than those of blood monocytes; 2) The phagocytosis activity and intracellular production of TNF after ex vivo LPS stimulation of alveolar monocytes were decreased in ARDS patients; and 3) Alveolar monocytes phagocytosis capacity and LPS-induced intracellular TNF concentrations did not differ according to their HLA-DR- vs HLA-DR+ phenotype.”
C12: Results regarding phagocytosis capacity are contradictory and should be better discussed. On the one hand, the authors suggest that phagocytosis activity of alveolar monocytes was decreased compared to circulating monocytes only in ARDS patients. On the other hand, they state (lines 459-460) « when compared with non ARDS-patients the phagocytic capacity of monocytes was preserved ». Further (line 471), they state : the expression of phagocytosis activity in MFI might be difficult to interpret (…). What do the authors mean ? Is the lower phagocytic activity (in MFI) in alveolar monocytes (compared to circulating) to be interpreted with caution ? However, the authors present it as one of the main results (lines 416-418). This discrepancy and this confusing message needs to be better explained.
R12: We thank the reviewer for raising this point, which indeed requires to be made clearer. The revised Discussion section of the manuscript (page 16, last paragraph) was modified, as follows:
“The phagocytic capacity of monocytes was not different in ARDS patients than in controls, contrasting with the down-regulation of HLA-DR expression on both alveolar and circulating monocytes, reflecting a defect in the antigen-presenting capacity. This finding was already reported in circulating monocytes from septic patients [38]. Such monocyte pattern resembles that described in alveolar monocytes from cystic fibrosis patients under the effects of pro-resolving mediators, which present enhanced phagocytic activity without evoking pro-inflammatory responses [15,39]. In ARDS patients, no difference was observed between alveolar and circulating monocytes when the phagocytosis capacity was quantified in percentage of positive cells, In contrast, using MFI we observed a lower phagocytic capacity of alveolar monocytes, which persisted after ex vivo LPS stimulation. It has been previously shown that functional, endotoxin-tolerant monocytes exhibit an increased phagocytic ability coupled with a conserved capacity to kill internalized pathogens, albeit with an impaired antigen presentation capacity [15,39,40]. Our data suggest that in pneumonia related-ARDS patients, the phagocytosis of alveolar monocytes seems to be less efficient than that of circulating monocytes, but as efficient as that of non-ARDS controls.”
C13 : The authors should be more cautious in their conclusions (lines 514-516 : this study showed that 514 monocytes have a deactivated status and an impaired inflammatory cytokine production 515 capacity but keep potent phagocytic activity). This is based on limited observations on very few patients and is to be considered as hypothesis-generating. Again, I remain confused about the message regarding phagocytic activity (« keep potent phagocytic activity »).
R12: The conclusion has been rephrased as follows:
“In conclusion, in patients with pneumonia-related ARDS, this study showed that monocytes have a deactivated status and an impaired inflammatory cytokine production capacity but keep similar phagocytic activity than non-ARDS controls…. “
C13 : In the introduction (lines 50-51), «Pulmonary infections account for the vast majority of ARDS risk factors » is probably overstated, as it is generally admitted that pneumonia accounts for 40-50% of ARDS etiologies. This sentence should be rephrased.
R13: We have downplayed this statement, as follows: “Pulmonary infections account for the majority of ARDS risk factors”.
Reviewer 3 Report
General Comments:
This study aimed to determine whether alveolar monocyte HLA-DR expression in patients with severe lung infection is associated with their phagocytic function. It is interesting and important for clinical physician.
Specific Comments:
- In Introduction section, authors state that down regulation of HLA-DR on circulating monocytes was consistently associated with an increased risk of nosocomial infections and a higher risk of death. However, it is more reliable that no recovery from serial circulating monocyte HLA-DR expression was associated with the high mortality in patients with sepsis (Crit Care 2011, 15(5):R224; Crit Care 2011, 15(5):R220).
- The major limitation in this work is very low case number as stated in Discussion section. Authors should carefully interpretate their results in Conclusion section.
Author Response
C1 : General Comments: This study aimed to determine whether alveolar monocyte HLA-DR expression in patients with severe lung infection is associated with their phagocytic function. It is interesting and important for clinical physician.
R1: We thank the reviewer for their positive comments on our work as well as for the constructive suggestions made.
C2. In Introduction section, authors state that down regulation of HLA-DR on circulating monocytes was consistently associated with an increased risk of nosocomial infections and a higher risk of death. However, it is more reliable that no recovery from serial circulating monocyte HLA-DR expression was associated with the high mortality in patients with sepsis (Crit Care 2011, 15(5):R224; Crit Care 2011, 15(5):R220).
R2. We thank the reviewer for raising this important information. These two references were added In the introduction section of the revised manuscript.
“During sepsis, a sustained decreased expression of the antigen presenting molecule human leucocyte antigen-DR (HLA-DR) on circulating monocytes is used as a surrogate marker of immune failure and higher risk of death [7–9]”
C3. The major limitation in this work is very low case number as stated in Discussion section. Authors should carefully interpret their results in Conclusion section.
R3. Another reviewer also rightly pointed out that our conclusions rely on a limited number of observations and should therefore be considered hypothesis-generating, which we completely agree with. The conclusion of the manuscript was rephrased, as follows:
“In conclusion, in patients with pneumonia-related ARDS, this study showed that monocytes have a deactivated status and an impaired inflammatory cytokine production capacity but keep similar phagocytic activity than non-ARDS controls. “
Round 2
Reviewer 2 Report
The authors succesfully addressed my comments and questions. The manuscript is now significantly improved and, in my opinion, can be accepted in its present form.
Reviewer 3 Report
No comments